# Allocating limited surveillance effort for outbreak detection of endemic foot and mouth disease

Ariel Greiner[1,2]*, José L. Herrera-Diestra[3], Michael Tildesley[4], Katriona Shea[1], Matthew Ferrari[1]

1 Center for Infectious Disease Dynamics and Department of Biology, Pennsylvania State University, University Park, Pennsylvania, United States of America, 2 Department of Biology, University of Oxford, Oxford, United Kingdom, 3 Department of Integrative Biology, The University of Texas at Austin, Austin, Texas, United States of America, 4 Zeeman Institute for Systems Biology & Infectious Disease Epidemiology Research, Mathematics Institute and School of Life Sciences, University of Warwick, Coventry, United Kingdom

* greinerariel@gmail.com

## Abstract

Foot and Mouth Disease (FMD) affects cloven-hoofed animals globally and has become a major economic burden for many countries around the world. Countries that have had recent FMD outbreaks are prohibited from exporting most meat products; this has major economic consequences for farmers in those countries, particularly farmers that experience outbreaks or are near outbreaks. Reducing the number of FMD outbreaks in countries where the disease is endemic is an important challenge that could drastically improve the livelihoods of millions of people. As a result, significant effort is expended on surveillance; but there is a concern that uninformative surveillance strategies may waste resources that could be better used on control management. Rapid detection through sentinel surveillance may be a useful tool to reduce the scale and burden of outbreaks. In this study, we use an extensive outbreak and cattle shipment network dataset from the Republic of Türkiye to retrospectively test three possible strategies for sentinel surveillance allocation in countries with endemic FMD and minimal existing FMD surveillance infrastructure that differ in their data requirements: ranging from low to high data needs, we allocate limited surveillance to [1] farms that frequently send and receive shipments of animals (Network Connectivity), [2] farms near other farms with past outbreaks (Spatial Proximity) and [3] farms that receive many shipments from other farms with past outbreaks (Network Proximity). We determine that all of these surveillance methods find a similar number of outbreaks – 2-4.5 times more outbreaks than were detected by surveying farms at random. On average across surveillance efforts, the Network Proximity and Network Connectivity methods each find a similar number of outbreaks and the Spatial Proximity method always finds the fewest outbreaks. Since the Network Proximity method does not outperform the other methods, these results indicate that incorporating both cattle shipment data and outbreak data provides only marginal benefit over the less

**Data availability statement:** The source code and data used to produce the results and analyses presented in this manuscript are available at https://github.com/ArielGreiner/FMDLimitedSurveillance/ and at https://figshare.com/articles/dataset/networks_2monthsep_rds/28429379?-file=52409981. All data needed to run the analyses described in this study have been provided except for the spatial location data required to reproduce the results of the Spatial Proximity method. The authors do not have permission to share the spatial locations of epiunits in the Republic of Türkiye because of issues related to identifiability and security posed by sharing the spatial locations of the epiunits in conjunction with the dates of FMD outbreaks in those epiunits. Instead, the authors provide randomized locations for each epiunit in the dataset to allow the code for the Spatial Proximity method to be run. All data queries should be made by contacting the SAP Institute in the Republic of Türkiye (info.turkey@sap.com).

**Funding:** AG, KS, MF were supported by NSF-NIH-NIFA (National Science Foundation - National Institutes of Health - National Institute of Food and Agriculture) Ecology and Evolution of Infectious Disease (https://new.nsf.gov/funding/opportunities/ecology-evolution-infectious-diseases-eeid) award DEB 1911962. AG was also supported by a Natural Sciences and Engineering Research Council of Canada (NSERC) Postdoctoral Fellowship (https://www.nserc-crsng.gc.ca/students-etudiants/pd-np/pdf-bp_eng.asp). MT was funded by a Biotechnology and Biological Sciences Research Council (BBSRC, https://www.ukri.org/councils/bbsrc/) grant BB/T004312/1. The funders had no role in study design, data collection and analysis, decision to publish, or preparation of the manuscript.

**Competing interests:** The authors have declared that no competing interests exist.

data-intensive surveillance allocation methods for this objective. We also find that these methods all find more outbreaks when outbreaks are rare. This is encouraging, as early detection is critical for outbreak management. Overall, since the Spatial Proximity and Network Connectivity methods find a similar proportion of outbreaks, and are less data-intensive than the Network Proximity method, countries with endemic FMD whose resources are constrained could prioritize allocating sentinels based on whichever of those two methods requires less additional data collection.

## Author Summary

Foot and Mouth Disease (FMD) poses a significant economic burden in countries where it is endemic. Developing surveillance systems that are efficient at detecting outbreaks is essential for managing and mitigating its impact in these countries. In this study, we use detailed outbreak and cattle shipment data from the Republic of Türkiye as a retrospective case study of endemic FMD. We use these data to evaluate the effectiveness of three data-informed surveillance allocation methods across a range of surveillance effort: 1) searching farms that frequently send and receive shipments, 2) searching farms near other farms with outbreaks, and 3) searching farms that receive many shipments from other farms with past outbreaks. We find that all three data-informed methods find 2.5-4 times more outbreaks than non-data informed methods but similar numbers of outbreaks to each other, even though some methods used more data than others. From this, we conclude that countries with endemic FMD and limited surveillance resources should consider developing surveillance systems based on either outbreak data or cattle shipment network data, whichever data requires less effort to collect.

## Introduction

Foot and Mouth Disease (FMD) is an acute systemic vesicular disease in cloven-hoofed animals worldwide, particularly common in domesticated cattle, sheep and goats. Mortality from FMD is rare but productivity is often severely reduced [1]. In some countries like the Republic of Türkiye, Tanzania and India, FMD is continuously circulating (endemic FMD), while others are considered free of FMD [2]. FMD can present a significant economic burden on countries because meat export from endemic countries is severely restricted [3]. Livestock farmers in endemic countries risk severe loss of income, either due to reduced productivity of their livestock due to the disease or because of imposed control policies such as culling [4,5]. The negative effects of FMD are felt at the country-level as well, due to the trade barriers, costs of measures to prevent and control the disease and general reduction in food security [5].

FMD spreads to nearby farms through direct contact with infected animals, through fomites on surfaces (sharing of farm equipment, veterinarians or personnel) or by

wind (rare) [1]. It may also spread to farms that receive livestock from infected farms through the shipment network, as the livestock and the vehicles that transport them may carry and spread the disease [1]. Thus, the farms with the highest risk of an FMD outbreak are likely those that are near already infected farms or are farms that receive cattle from already infected farms.

In epidemic settings, FMD has been controlled and eliminated through a mixture of movement restrictions, targeted depopulation initiatives and targeted vaccination campaigns [6,7]. Movement restrictions limit or prevent the movement of livestock between farms and to markets, significantly reducing the rate of spread of FMD in a region. Movement restrictions are often recommended to be broad [8], but some have called for them to be more targeted to reduce excess loss of profit and livestock [9,10]. Targeted depopulation initiatives and vaccination campaigns focus on livestock in farms with FMD outbreaks and farms that are a certain distance away from farms with outbreaks (ring culling/vaccination) [6]. Targeted depopulation initiatives are often found to be the most effective and rely on the existence of effective surveillance programs to find outbreaks quickly [11–13].

The vast majority of FMD cases occur in countries where FMD is endemic [14–16], and designing effective control measures for such countries can reduce the economic burden and accelerate progress towards FMD-free status. The number of FMD outbreaks in countries with endemic FMD fluctuates over time–e.g., in the Republic of Türkiye there were large outbreaks of serotype A in 2006 and 2011 and serotype O in 2007 and 2010 [17]. Control measures in endemic countries have focused on reducing the prevalence and economic burden [5] of the disease in all or parts of the country through shipment restrictions and mass (entire regions) and ring vaccination campaigns [7,18]. In the Republic of Türkiye, where FMD is endemic, control measures aided by an extensive surveillance program managed to reduce the prevalence of the disease from 45% to 5% between 2008 and 2018 [19]. However, many countries with endemic FMD have less effective surveillance programs and thus are not able to target their control measures as effectively [7,20].

If one could monitor the farms with the highest risk of an FMD outbreak in countries with endemic FMD, it would be easier to manage at low prevalence (i.e., before any particular outbreak takes hold) [21]. This would then delimit the necessary scope of control measures and prevent FMD from spreading widely. Control measures are costly [20], so pursuing targeted control measures at low prevalence is both more productive and more cost effective. This has been demonstrated in FMD-free countries –Tildesley et al. [22] found that vaccinating near known infected farms would have reduced the size of the 2001 FMD epidemic in the UK and Schley et al.[9] found that, with sufficient information, targeted movement restrictions in the UK would have been effective and led to fewer cattle culled in the 2001 and 2007 epidemics.

Surveillance of farms will help detect outbreaks early, but as surveillance is expensive and time-consuming, sentinel surveillance strategies are often prioritized. Under a sentinel surveillance strategy, farms that are thought to be at high risk of infection are identified and asked to report outbreaks [23]. Thus, it is important to identify which farms should be preferentially allocated for sentinel surveillance in countries with endemic FMD so that early outbreaks can be discovered before FMD prevalence becomes too high. One method that can help find high-risk farms is to model a country as a network of nodes (farms) and edges (shipment pathways) along which diseases may transmit and then use network theory [24–27] and past outbreak data to determine which locations might be most at risk of outbreaks. Network theory has been used before to find high risk farms in FMD-free countries experiencing FMD outbreaks. For example, Dawson et al. [28] used animal movement data from the UK to determine that targeting nodes with the highest number of animal movements allowed them to accurately predict the size and spread of simulated FMD-like epidemics.

Much of the theory on how infectious processes spread on networks that informs our understanding of which nodes are highest risk assumes an epidemic setting [29]. It is less clear which nodes are highest risk for outbreaks in an endemic setting [17]. The importance of designing active and passive surveillance strategies is highlighted by EuFMD as a core competency for a country to reach phase 2 of the Progressive Control Pathway advocated by EuFMD for countries with endemic FMD [30]. In this study, we analyze retrospective FMD incidence in an endemic setting with a highly resolved network of livestock shipments and outbreak incidence to evaluate active surveillance strategies for efficient outbreak

detection. We compare the effectiveness, with respect to outbreak detection, of sentinel surveillance methods informed by shipment network and/or outbreak incidence information to help identify high risk nodes for FMD in endemic settings. By using retrospective, empirical data, rather than simulation we can assess the performance of outbreak detection under realistic spatial and temporal distributions [31].

For this study we use retrospective data (2007–2012) from the Republic of Türkiye, where FMD is endemic. During this time period there was an extensive passive surveillance system to help manage FMD outbreaks in domestic cattle. The Republic of Türkiye's surveillance program recorded the start date and location (epiunit; defined as a geographic area containing one or more farms) of all reported outbreaks and records which cattle were shipped between which epiunits at which time. Guyver-Fletcher et al. [32] used these data to show that shipment-mediated persistence of FMD was possible and Herrera-Diestra et al. [17] used these data to show that epiunits that were more central in the network had a higher risk of an outbreak – demonstrating the usefulness of network analysis for assessing the persistence of FMD and for identifying high risk epiunits in the Republic of Türkiye.

In this study, we use these highly resolved, retrospective data on FMD outbreaks in an endemic setting as a case-study for assessing the effectiveness of sentinel surveillance strategies based on [1] network properties, [2] spatial proximity to outbreaks, and [3] network proximity to outbreaks. We assess how each of these surveillance schemes, with varying levels of effort, perform at detecting outbreaks from the Republic of Türkiye dataset as an illustration of the likely performance of these strategies in other endemic settings with minimal FMD surveillance infrastructure. We explore the value of using outbreak incidence data and shipment network data to allocate limited sentinel surveillance effort with the goal of providing some general guidelines for designing sentinel surveillance programs in countries with endemic FMD with minimal existing surveillance infrastructure. Better allocation of surveillance effort will help countries with endemic FMD better plan and utilize their resources – allowing for more efficient and less costly management protocols for controlling FMD outbreaks.

## Methods

### Overview

We used cattle shipment data and outbreak incidence data from January 2007 to July 2012 from the Republic of Türkiye to develop and compare general methods (Fig 1) for allocating limited sentinel surveillance effort for FMD outbreak detection in countries with endemic FMD and minimal existing FMD surveillance infrastructure. Using the cattle shipment data (edgelists with information about the origin, destination, timing and frequency of cattle shipments), we constructed overlapping two-month directed-weighted cattle shipment networks (nodes = epiunits, edges = cattle shipment routes) based on the networks calculated by Herrera-Diestra et al. [17] from the same dataset. We then used the outbreak incidence data, the geographic epiunit locations and these two-month networks to develop and compare three surveillance methods (Fig 1) across seven different levels of surveillance effort to investigate which method(s) would be the most effective at finding recorded outbreaks and if effectiveness varied by effort level.

### Model system

FMD is continuously circulating in the Republic of Türkiye and, in response, the government has developed an extensive surveillance program that monitors FMD outbreaks and tracks cattle movement among farms. For this study, we used data from the Turkish Veterinary authorities, facilitated by the European Commission for foot-and-mouth disease (EuFMD), who granted us access to data from the TurkVet database. The data for this study extended from January 1, 2007 to July 4, 2012 and covered 54,096 epiunits, 5,125 outbreaks and 14,261,447 cattle shipments. Epiunits are geographically defined regions of the Republic of Türkiye that contain comparable numbers of cattle to other epiunits [33]. Epiunits are considered the basic epidemiological units for recording FMD outbreaks in the Republic of Türkiye [34]. These data were collected using a passive surveillance system; farmers reported infected cattle with FMD-like symptoms and then these cattle

 

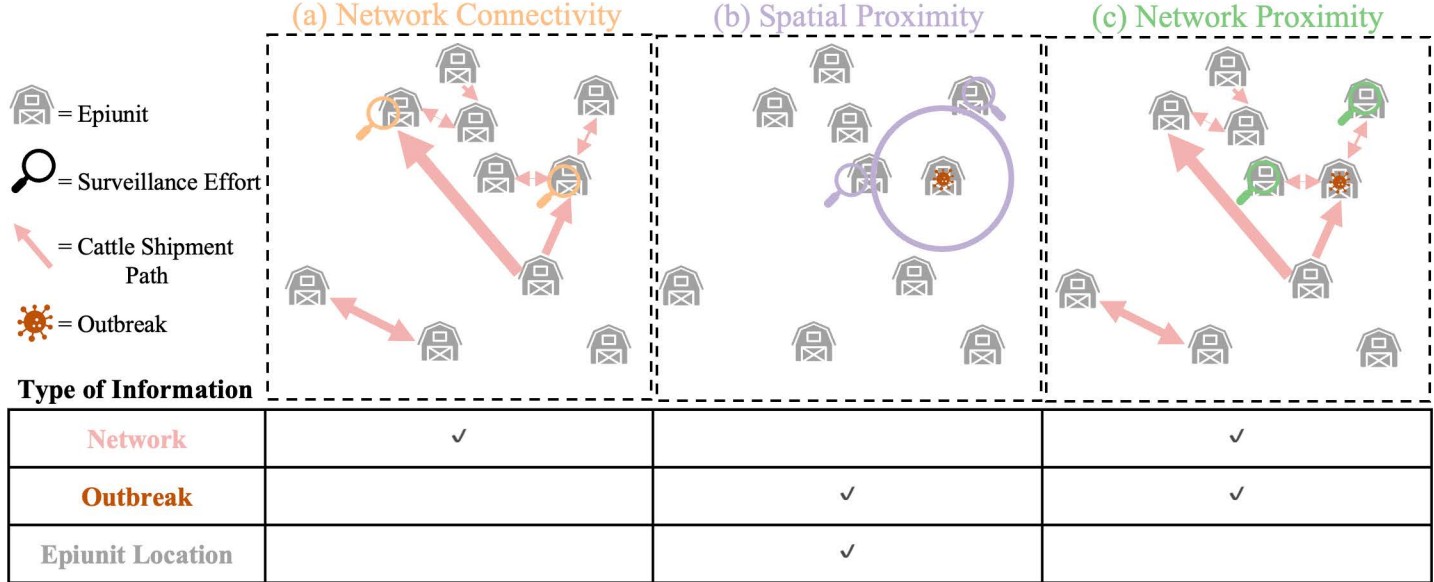

**Fig 1. Surveillance Methods Summary.** The three different data-informed surveillance methods are shown diagrammatically with an accompanying description of which types of information they are informed by. For each surveillance method, two magnifying glasses are shown on the epiunits that would have been selected by each surveillance method at a surveillance effort of 20% (2/10 epiunits). Epiunits with outbreaks on them in the diagram indicate epiunits with outbreaks with start dates in month $t$ (i.e., 'outbreak epiunits'). The circle in (b) indicates the search radius chosen at a surveillance effort of 20%. Note that 'Network' and 'Outbreak' information are dynamic and require consistent collection while 'Epiunit Location' information (the latitude and longitude of the centroid of the epiunit) is static and does not change over time. 'Outbreak' information is a record of the epiunit and start date of a particular outbreak. 'Network' information records the start and end epiunit of all cattle shipment events. Note that the 'Network Connectivity' surveillance method selects sites to survey without using the information in the outbreak dataset, but the outbreak dataset is used to assess whether the epiunits selected for surveillance by this method experienced an outbreak during the relevant time period (see the '*Network Connectivity Method*' subsection of the '*Surveillance Methods*' section for more details). Icons are from Microsoft Powerpoint and are free to use without royalty or copyright (https://support.microsoft.com/en-us/office/insert-icons-in-microsoft-365-e2459f17-3996-4795-996e-b9a13486fa79).

were tested for FMD. If herds were confirmed positive for FMD, viral serotyping was usually performed [33]. The data from this extensive surveillance program thus forms an ideal case study for assessing different sentinel surveillance allocation methodologies for countries with endemic FMD which currently have limited existing surveillance infrastructure. Here, we treat the highly resolved list of outbreaks from the TurkVet database as the 'truth' and evaluate how different allocation schemes, over varying levels of effort, perform at detecting the recorded outbreaks.

This dataset describes when an outbreak was reported in a particular epiunit and thus we define an outbreak at the epiunit scale. In the context of this study, we define an outbreak as starting as soon as one or more cattle in an epiunit is reported to be infected with FMD. For each outbreak, this dataset contains information on which epiunit experienced the outbreak, the start date of the outbreak and the serotype of the outbreak. A and O serotypes of FMD are most common in this dataset and have been continuously endemic in the Republic of Türkiye, while Asia-1 re-emerged in 2011 after being unobserved since 2001 [17]. In our dataset, we observed 1479 serotype A outbreaks, 1784 serotype O outbreaks, 455 Asia 1 outbreaks while 1458 outbreaks did not have serotype information.

The original dataset contained 55,193 rows of epiunits, but 1097 of those rows were removed because they were empty or contained duplicate epiunits. In the original dataset, only 40,746 epiunits had a unique set of coordinates, 15,445 epiunits shared 2,095 coordinates. To avoid removing all of these epiunits from the dataset, approximate coordinates were found for 2,445 epiunits through geocoding using the Google Maps Application Programming Interface and coordinates for the remaining 13,000 epiunits were chosen through restricted sampling from a uniform distribution (to ensure that the epiunits remained in the correct district, as the district of each epiunit was known) (additional details in [33]).

## Network construction

FMD has an incubation time of 2–14 days [35] and a serial interval of 8–9 days at farm level (symptom onset at farm 1 to symptom onset at farm 2) [36], thus the period between exposure at one farm to symptom onset at another farm could be as long as twenty days. With that in mind, we constructed directed-weighted networks from two-month overlapping time periods ((month $t$, month $t+1$) - i.e., network 1: month 1 + month 2, network 2: month 2 + month 3, etc.; $t$: [1,67]) of cattle shipment data (Table A in S1 Text for more information on the characteristics of these two-month networks). The two-month time period allows us to capture the start of any outbreaks in month $t+1$ that spread from, or were the result of, outbreaks in month $t$, with allowance for uncertainty in exact start date of either outbreak. Thus, the time period of two months allowed us to incorporate all of the network connections along which an outbreak that started in month $t$ could have spread to other epiunits. We look at overlapping two-month networks so outbreaks in every possible month $t$ and month $t+1$ (within the time period, i.e., January 2007 is never a month $t+1$ and July 2012 is never a month $t$) are considered as both potential start and end points of transmission.

We constructed these networks by summing up all of the cattle shipment events between every ordered pair of epiunits $i$ and $j$ for each two-month time period (i.e., $(i,j)$ =/= $(j,i)$ where $(i,j)$ represents a shipment of cattle from epiunit $i$ to epiunit $j$). The weight of each edge in the network describes the frequency of cattle shipments from epiunit $i$ to epiunit $j$ in that two-month time period. Note that not all epiunits sent or received shipments of cattle in every two-month time period (indeed, some did not report cattle shipments throughout the entire January 2007 to July 2012 period) and so these networks varied in size.

## Surveillance methods

We tested three different surveillance methods (Network Connectivity, Spatial Proximity, Network Proximity) that were all informed by different data (network information, epiunit location or outbreak information) (Fig 1) from the Republic of Türkiye, hence we will refer to these as 'Data-Informed Surveillance Methods'. In Fig 1 we show a simple example of how we would select the best two epiunits (out of the ten shown); using the Network Connectivity method we would select the two most highly connected epiunits with no consideration for outbreak location or start date (Fig 1a), with the Spatial Proximity method we would select the two epiunits in closest spatial proximity to an epiunit with an outbreak at that time (Fig 1b) and with the Network Proximity method we would select the two epiunits that an epiunit with an outbreak at that time sends cattle to most frequently (Fig 1c). We chose to compare these three surveillance methods as they made adequate use of the data at hand, were all relatively straightforward to implement, relate to standard reactive control measures for FMD [6,7,11,12,18] and have been proposed as methods for allocating sentinel surveillance locations in the past for FMD and other diseases [28,37–39]. Note that the sentinel surveillance methods that we discuss here are 'active' surveillance strategies, but the Network Proximity and Spatial Proximity surveillance allocation methods rely on some prior collection of outbreak data, which could have been through passive or active surveillance methods. We also calculated the effectiveness of searching randomly-selected epiunits for outbreaks ('Random Method') in order to compare the data-informed surveillance methods against a non-data informed surveillance method.

For each surveillance method we explored 7 different levels of surveillance effort (5%, 10%, 15%, 20%, 25%, 30%, 35%; which correspond to 2704, 5409, 8114, 10819, 13524, 16228, 18933 epiunits respectively). We did not survey beyond 35% because the Spatial Proximity method was unable to survey all months at 35% effort level (as explained in the 'Spatial Proximity Method' section below); however, we do include the results for surveying at 40%-100% surveillance effort level in the supplementary material (Fig A in S1 Text). The sentinel surveillance epiunits allocated by the Spatial Proximity method, the Network Proximity method and the Network Connectivity method were assessed based on whether they experienced outbreaks in the second ($t+1$) month of each two-month overlapping network; thus the sentinel

surveillance sites determined by these three methods were dynamic in nature as they were allowed to change month-to-month. The sentinel surveillance sites allocated by the All-Month Network Connectivity method (defined below) and the Random method were static as they were only calculated once for the entire study time period. A surveillance effort level of $x$ means that we surveyed approximately $x$% of all epiunits in the dataset using that surveillance method (not just those found in that particular two-month network; Fig B in S1 Text). The specifics of how $x$% of epiunits were selected is described below for each method.

**Random method.** As a null model, we selected $x$% of epiunits at random 10,000 times ($x$ = surveillance effort level) and assessed how many of those epiunits had an outbreak at any point in our dataset, and then recorded the mean value across the 10,000 replicates. This surveillance method was not informed by geographic location data, outbreak data or cattle shipment network data.

**Network connectivity method.** Since FMD may transmit via the cattle shipment network in the Republic of Türkiye, we first decided to survey the epiunits that were deemed most central in the network (Fig 1a), as these experience the most cattle shipments and thus would likely be at higher risk of an outbreak [24–27]. We chose 'degree' (number of nodes connected to a focal node in the network) as our connectivity metric because [1] it is simple to calculate and understand, [2] it forms the basis of most other centrality metrics (betweenness centrality, eigenvector centrality, strength, etc.; [40,41]) and [3] because we found that it performed similarly to all of the other connectivity metrics we tested, even those that used more information from the two-month networks (Fig C in S1 Text).

To calculate the degree of each epiunit, we computed the number of different epiunits that every epiunit sent to or received cattle from during the two months in total, as summarized by each two-month overlapping network. We calculated the degree of every node in every two-month overlapping network and ranked the epiunits from highest degree to lowest degree (epiunits with the same degree were ordered indiscriminately by taking the default ordering from igraph [42] (Fig 1a)). For each surveillance effort level $x$, we assessed how many of the $x$% top ranking epiunits had an outbreak (as recorded in the outbreak dataset) that started in month $t+1$ of each two-month time period.

Herrera-Diestra et al. [17] determined that epiunits that experienced outbreaks were more central in the full cattle shipment network that described all of the cattle shipments from January 2007 to July 2012. Thus, we additionally calculated the degree of each epiunit in the network that described all of the cattle shipments from January 2007 to July 2012 ('All-Month Network Connectivity Method'). We then quantified the proportion of outbreaks (that occurred at any time in our outbreak dataset) detected in the $x$% of epiunits with the highest degree in this network, for each surveillance effort level $x$. We note that the All-Month Network Connectivity method includes cattle shipments, and thus edges, that may occur after a given outbreak. Though unrealistic in practice, because it uses future information, this is consistent with other studies [43,44] that generate a static connectivity matrix by integrating over moves over a long time window. We present this analysis in the supplementary material (Fig D in S1 Text) for comparison because we were interested in assessing the relevance of using time-sensitive shipment information as compared to general shipment information for a region.

**Spatial proximity method.** The second data-informed surveillance method assessed the number of outbreaks that could be detected through searching epiunits that were geographically close to epiunits that had experienced recent outbreaks (Fig 1b). This is a reasonable surveillance method to explore as FMD is known to transmit between nearby farms through direct contact with infected animals or through fomites (i.e., passed along through the sharing of equipment, sharing of veterinarians or face-to-face interactions among humans and cattle in neighbouring epiunits) and many control methods are based on this known transmission pathway [6,11,12,13].

To search epiunits near epiunits with recent outbreaks, we recorded how many outbreaks with start dates in month $t+1$ occurred in epiunits that were geographically close (within a certain radius) to epiunits with outbreaks with start dates in month $t$ (outbreak epiunit) (Fig 1b). For a given two-month network and level of surveillance effort, $x$, the radius chosen was the smallest radius away from each outbreak epiunit that, when summed across all of the outbreak epiunits, included

at least *x*% of all epiunits ((Fig 1b); (Fig E in S1 Text) shows the radius sizes used for each surveillance effort level for each month *t + 1*, which varied from 4.44km to 463.98 km). For each two-month network and surveillance effort level *x*, we increased the radius in increments of 1/50 degree units until it satisfied this condition. For every surveillance effort level and every two-month network, one radius was used to search around every outbreak epiunit.

Thus, if one month *t* had more outbreak epiunits than another month *t*, the radius chosen would likely be smaller for the first month *t* than for the second at the same surveillance effort level (Fig E in S1 Text). This method ensured that we could survey a comparable number of epiunits across months and across surveillance methods (Fig B in S1 Text for more details). This method only used the outbreak data and the geographic locations of the epiunits and did not account for the existence of network connections among epiunits.

**Network proximity method.** The final data-informed surveillance method assessed how many outbreaks could be detected through searching epiunits that were closest, in network-space (calculated from the two-month networks), to epiunits that had experienced outbreaks with start dates in month *t* (outbreak epiunits, Fig 1c). We define the closest in network-space epiunits as those epiunits that receive the most cattle shipments directly from outbreak epiunits.

For every outbreak epiunit, we ranked the epiunits they sent cattle directly to (Level 1 epiunits, i.e., first order neighbours) based on the weight of the edges that connected them to the outbreak epiunit (i.e., number of shipment events from the outbreak epiunit to each level 1 epiunit; edges with more events have higher weight, ranked highest to lowest). If two such epiunits had the same weight, their order was determined indiscriminately by taking the default ordering from igraph [42]. Once all of the Level 1 epiunits were considered, the epiunits connected to the Level 1 epiunits (i.e., the Level 2 epiunits or second order neighbours) were ranked in the same fashion. Additional levels of epiunits were considered as necessary for each surveillance effort level *x*.

For every surveillance effort level and every two-month network, one rank and level combination was used to search around every outbreak epiunit. The rank and level combination chosen was the smallest rank and level combination away from each outbreak epiunit that, when summed across all of the outbreak epiunits, included at least *x*% of all epiunits (Fig 1c). Thus, if one month *t* had more outbreak epiunits than another month *t*, the rank and level combination chosen would likely be smaller for the first month *t* than for the second at the same surveillance effort level. We then assessed how many of the top *x*% of epiunits had an outbreak that started in month *t + 1* of that two-month time period.

This method again ensured that we could survey a comparable number of epiunits across months and across surveillance methods (Fig B in S1 Text for more details). This method used both the outbreak data and the full weighted, directed cattle shipment networks. Note that for certain two-month networks, less than 35% of the total epiunits in the network could be reached from the outbreak epiunits. Thus at the 35% surveillance effort level, for this surveillance method, there are some months that do not record a 'number of outbreaks detected' and the average across months is N/A (Figs F and G in S1 Text).

### Outbreak detection

We calculated the number of outbreaks detected in month *t + 1* (at every month *t + 1*, *t + 1*:[2,67]) using each of these surveillance methods. Note that when we refer to the number or percentage of outbreaks detected in a 'month', we are referring to the number detected in month *t + 1* of the (month *t*, month *t + 1*) two-month network. We also calculated the average number of outbreaks detected by each method at each surveillance effort level, across all months for which we had network and outbreak data (January 2007 to July 2012). Additionally, we assessed the number of outbreaks of serotype O or serotype A detected by serotype-specific versions of each of these surveillance methods, but since the serotype-specific results were similar to the non-serotype-specific results we present those results in the supplementary material (Tables C and D in S1 Text).

## Results

### Across months

All three data-informed surveillance methods (Fig 1) detect similar percentages of outbreaks at each surveillance effort level but at higher surveillance effort levels the Network Proximity method and Network Connectivity method detect 5–10% more outbreaks than the Spatial Proximity method, averaged across time (January 2007 to July 2012, Fig 2). At the lowest surveillance effort level (5%), the data-informed surveillance methods detect four to five times more outbreaks on average than were detected by the random surveillance method (Fig 2). As surveillance effort level is increased, the data-informed surveillance methods detect relatively fewer outbreaks. At 30% surveillance effort level these methods only detect approximately two times more outbreaks on average than the random surveillance method (Fig 2). Across surveillance effort levels,

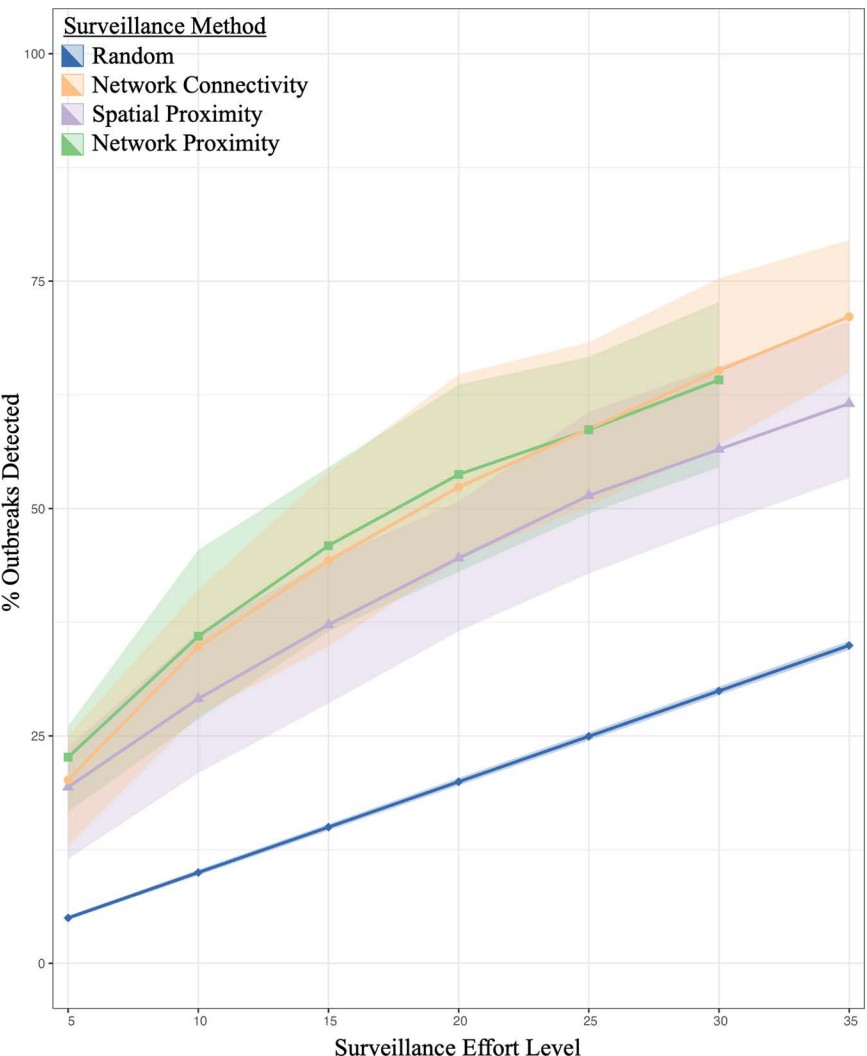

**Fig 2. Performance of Surveillance Methods Across Surveillance Effort Levels.** The points represent the average percent of outbreaks each surveillance method detected across all *t + 1* months at each surveillance effort level (5-35%). The ribbons represent the variability across months (the interquartile range). The Network Proximity method is unable to survey over 30% of the epiunits for all of the two-month networks and so it is only plotted until the 30% surveillance effort level (as opposed to 35%).

the Spatial Proximity method consistently finds the fewest outbreaks of the three data-informed surveillance methods on average (Fig 2). The Network Proximity method and Network Connectivity method find a similar percentage of outbreaks across surveillance effort levels on average (Fig 2). Beyond 35% surveillance effort level, the relative performance of the different surveillance methods (that were able to survey beyond 35%) did not change until >60% when the Network Connectivity method became restricted by the number of epiunits present in the two-month networks (Fig A in S1 Text).

The Network Connectivity surveillance method detected more outbreaks on average across months than the All-Month Network Connectivity surveillance method, across all surveillance effort levels (Fig D in S1 Text). The All-Month Network Connectivity surveillance method finds the smallest percentage of outbreaks at low surveillance effort level, but the second most (on average) at the highest surveillance effort level (Fig D in S1 Text).

When we used the data-informed surveillance methods to search for only serotype A or serotype O outbreaks, they detected a similar percentage of outbreaks averaged across months as the non-serotype specific surveillance methods (Tables C and D in S1 Text). In general, the Network Connectivity method was the best at finding serotype A outbreaks on average and the Network Proximity method was the best at finding serotype O outbreaks on average but the differences in surveillance method performance are fairly minor (Tables C and D in S1 Text).

### Month by month

As surveillance effort level was increased, the three different data-informed surveillance methods all detected more outbreaks per month (Fig 3 and Fig F in S1 Text). As the number of outbreaks in a month decreased, the proportion of outbreaks detected generally increased for all surveillance methods and across surveillance effort levels (Figs 3, 4 and Figs F and G in S1 Text). For example, when there were fewer than 15 outbreaks in a month, the surveillance methods detected up to 67% of outbreaks at a 5% surveillance effort level (Fig 3a) and up to 100% of outbreaks at a 35% surveillance effort level (Fig Fd in S1 Text) (across methods). In contrast, when there were more than 50 outbreaks in a month, surveillance only detected up to 44% of outbreaks at a 5% surveillance effort level (Figs 3a and 4a) and up to 89% of outbreaks at a 35% surveillance effort level (Figs Fd and Gd in S1 Text) (across methods). As the number of outbreaks increased, the spatial radius required to perform the Spatial Proximity method at the appropriate surveillance effort level decreased (and the radius increased as surveillance effort level increased; Fig E in S1 Text). In general, the three data-informed surveillance methods all showed fairly similar outbreak detection patterns to each other (Fig 3 and Fig F in S1 Text).

Even though on average across months the Network Proximity method and Network Connectivity method consistently detected more outbreaks than the Spatial Proximity Method (Fig 2), none of the data-informed surveillance methods consistently performed better than the other two for all months across surveillance effort levels (Fig 4 and Fig G in S1 Text). Also, no method was consistently better at detecting outbreaks as the number of outbreaks per month decreased (Fig 4 and Fig G in S1 Text). As surveillance effort level increased, the Network Connectivity method performed better in more months compared to the Network Proximity method while the Spatial Proximity method performed better in the fewest months at every surveillance effort level (Fig 4 and Fig G in S1 Text).

### Discussion

All data-informed surveillance methods detected 2 to 4.5 times more outbreaks than the random surveillance method. For example, at a 5% surveillance effort level the data-informed methods detected roughly 20% of the outbreaks (on average), while the random surveillance method only detected 20% of outbreaks when searching at a 20% surveillance effort level (Fig 2). None of the data-informed sentinel surveillance methods were consistently better across surveillance effort levels or across months than any other method (Figs 2–4). However, we determined that surveillance methods informed by network data (Network Connectivity, Network Proximity; Fig 1) performed better than the Spatial Proximity method on average across all surveillance effort levels (Fig 2; especially at high surveillance effort levels) and across months (Fig 4 and Fig G in S1 Text). As surveillance effort level increased, the surveillance method that relied exclusively on network

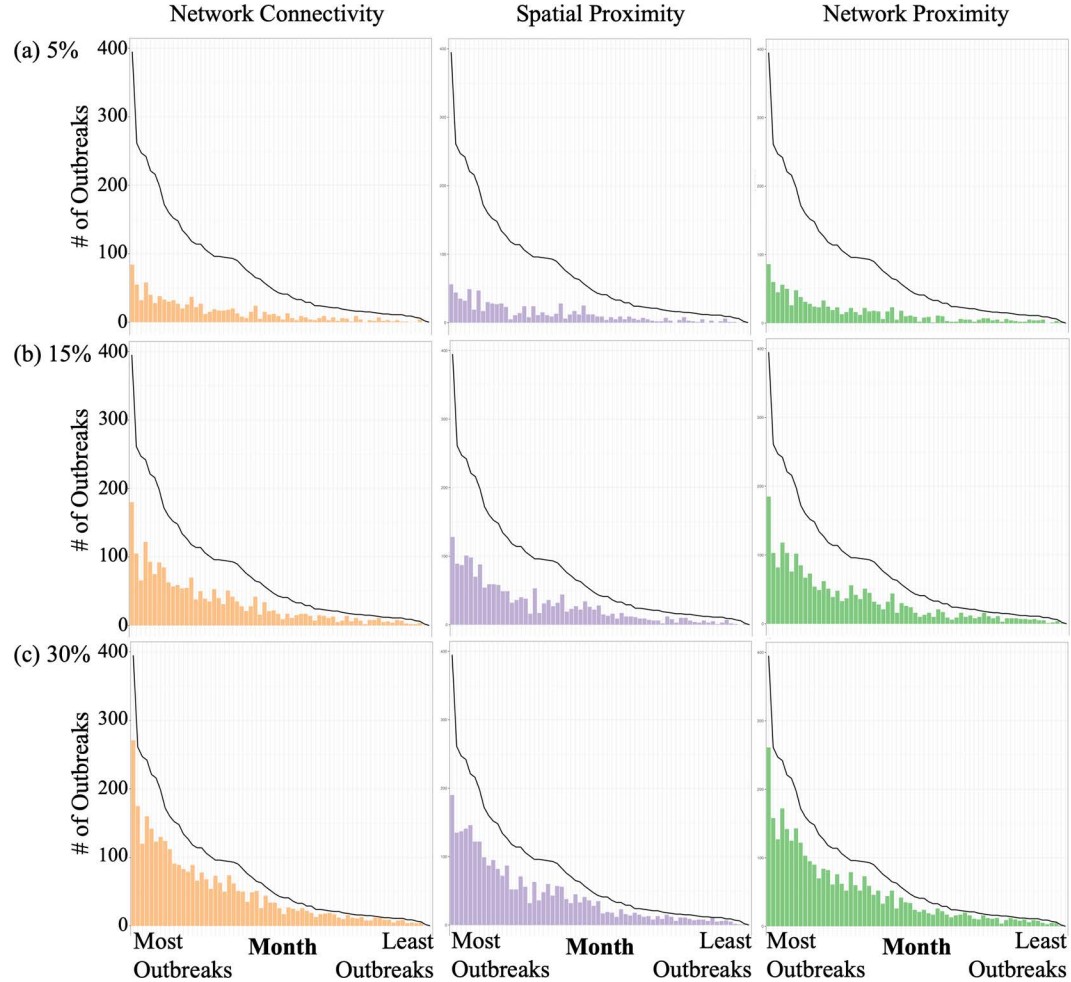

**Fig 3. Month-by-Month Surveillance Method Performance.** The bars in each panel show the number of outbreaks detected by each surveillance method at each surveillance effort level (5%, 15% 30% respectively) at each *t* + *1* month. The black line shows the number of outbreaks reported in each *t* + *1* month. The *t* + *1* months are ordered by decreasing number of outbreaks, with the left-most month having the most outbreaks. Versions of these graphs that correspond to 10%, 20%, 25% and 35% surveillance effort levels are in the supplementary material (Fig F in S1 Text).

properties (Network Connectivity method) improved in comparison to the other methods (Fig 2). Gilbert et al. [45] determined that the spread of FMD in the Republic of Türkiye had become more associated with long-range transportation of animals than with short-range transmission; a finding that these results support. The surveillance method that used both network and outbreak information ('Network Proximity'; Fig 1) was the most effective (on average) at small surveillance effort levels, but at larger surveillance effort levels it was not much better than the less informed methods and it also became limited by the availability of edges in the network (at 35% surveillance effort level; Figs Fd and Gd in S1 Text). These results imply that knowing where and when outbreaks happen is less crucial when more epiunits are under surveillance (Fig 2). Lastly, we determined that all of the data-informed surveillance methods detected a greater proportion of outbreaks as the number of outbreaks per month decreased; even the Network Connectivity method, whose allocation method was not influenced by the number of outbreaks in a month, showed this pattern. For example, when we used the data-informed surveillance methods to survey 5% of epiunits, they detected over 60% of outbreaks when outbreaks were rare (fewer than 15 outbreaks per month) (Figs 3 and 4a).

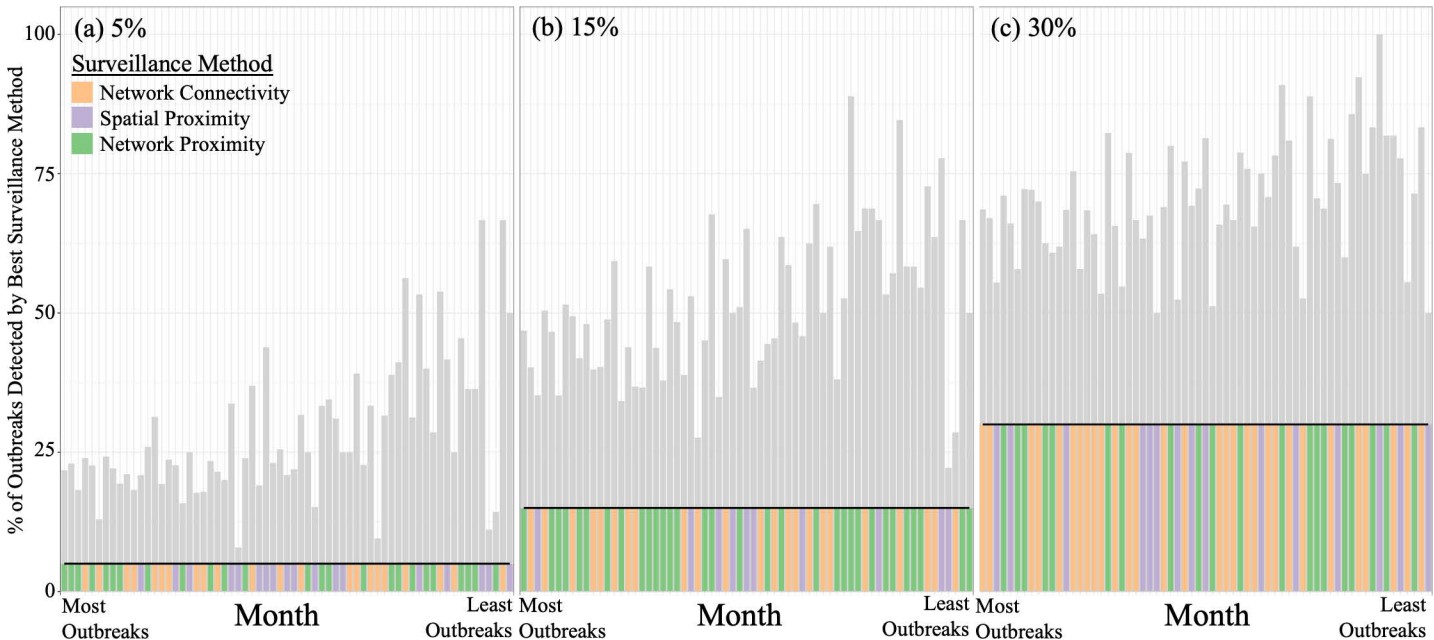

**Fig 4. Best Surveillance Method Each Month.** Each panel shows the percent of outbreaks detected by the data-informed surveillance method that detected the most outbreaks in each $t+1$ month at each surveillance effort level. The bars indicate the percent of outbreaks detected by the best surveillance method and the horizontal black line indicates the percent of outbreaks detected by the Random surveillance method at that surveillance effort level. The coloured bars indicate the data-informed surveillance method that detected the most outbreaks at that surveillance effort level for that $t+1$ month. The $t+1$ months are ordered by declining number of outbreaks, with the left-most month having the most outbreaks. Versions of these graphs that correspond to 10%, 20%, 25% and 35% surveillance effort levels are in the supplementary material (Fig G in S1 Text).

We also found that restricting our analysis to individual serotypes did not improve our ability to find outbreaks, as we would expect if we were identifying explicit chains of transmission. For example, sentinel surveillance methods that searched for serotype O FMD outbreaks detected slightly fewer outbreaks overall than serotype A specific surveillance methods and the non-serotype specific surveillance methods (Tables C and D in S1 Text). This result implies that collecting serotype-specific information does not improve our ability to allocate sentinel surveillance locations and that informed sentinel allocation decisions could be made from non-serotype-specific information alone.

Additionally, we determined that collecting time-specific shipment network information improved our ability to find outbreaks. When we used the cattle shipment network calculated from all of the cattle shipment data (January 2007 to July 2012), the All-Month Network Connectivity surveillance method detected fewer outbreaks than the average number detected by the Network Connectivity surveillance method informed by only two months of data (Fig D in S1 Text; also holds for the serotype-specific methods– Tables C and D in S1 Text). This is likely because the two-month cattle shipment Network Connectivity surveillance method is only informed by cattle shipments that happened around the date of the potential outbreaks and is not affected by additional shipments that happened far before or far after the outbreaks being searched for. Operationally, this implies that countries with endemic FMD should prioritize collecting shipment data from around the time of the outbreak start date and should be wary of using shipment data that is from far before or far after an outbreak as it may be misleading at worst, or exhibit diminishing returns at best.

The results of this study were calculated from retrospective data from the Republic of Türkiye, and it is important to acknowledge that farming demography, animal movement networks and available resources for FMD surveillance to incentivize participation, including programmatic support from relevant industry and governmental stakeholders, may differ in other countries with endemic FMD. However, we can still extrapolate some general suggestions for designing

sentinel surveillance programs (across a variety of surveillance effort levels) to detect future FMD outbreaks in countries with endemic FMD that have limited resources. Since surveillance methods informed by network data detected more outbreaks at high surveillance effort levels but roughly the same number of outbreaks as other methods at a low surveillance effort level (on average), our results would suggest that countries with endemic FMD and limited surveillance resources should prioritize collecting whichever of those data sources would require the least additional resources, while countries with endemic FMD and more resources available for surveillance might prioritize collecting network data (as, at higher surveillance efforts, these methods detect 5–10% more outbreaks which could be quite significant, practically speaking). We phrase it this way so as to not make assumptions about which types of data are already being collected by different countries (i.e., the existing surveillance infrastructure), but we do acknowledge that animal movement data is less likely to be available in low resource settings (note that India, another country with endemic FMD, has created a livestock movement database [46]). Note that Stage 1 of the Progressive Control Pathway for FMD developed by FAO and EuFMD [30] encourages building systems to describe and document outbreaks and cattle movement, thus acknowledging the importance of these two types of data. In the Republic of Türkiye, the mean shipment distance was fairly short (~5km, but note that some were >120km; [17]) which may be why the network-informed methods performed similarly to the Spatial Proximity method (Fig 2). If farms are further apart on average, network information may become even more important as there would likely be less transmission between neighbouring farms. Our results also indicate that dynamic sentinels might be ideal; we determined that the best sentinel surveillance methods were sensitive to time-specific network and outbreak information and thus the best sentinel surveillance locations changed over time. Since all of the surveillance methods detected a larger proportion of outbreaks when outbreaks were rare (Figs 2 and 3; including the 'Network Connectivity' method which did not use any outbreak information), it implies that outbreaks may be easier to predict and track using data-informed methods in countries with endemic FMD when they are rare, which is promising as it is often most important to find outbreaks when there are fewer of them in order to better control the disease.

Past studies of sentinel surveillance allocation for FMD and other diseases that used network data tended to focus on allocating sentinels through a Network Connectivity-type method, and found similar results. Dawson et al. [28] used livestock transport network data from the UK to determine that using a Network Connectivity-type method for surveillance allocation finds more simulated FMD-like outbreaks than various random allocation methods. Interestingly, in Dawson et al. [28]'s simulated non-endemic setting, the top 20% of nodes detected through the network connectivity-type method are sufficient to predict epidemic size to within 90% confidence; in our study, however, the top 20% of epiunits detected using the Network Connectivity method only encompassed approximately 40% of the outbreaks. This implies that Network Connectivity-type methods may perform better in simulated epidemic settings (which have perfect observation of the network and the outbreaks) than it did for realized endemic FMD outbreaks. Frössling et al. [38] combined serological survey results and cattle movement data to compare a Network Connectivity-type method and a Network Proximity-type method for allocating herds to survey for detecting BRSV (bovine respiratory syncytial virus) and BCV (bovine coronavirus) infections, also finding that surveillance methods informed by network data from the relevant time period found more infections than random methods (also finding that the Network Connectivity-type method performed slightly better than the Network Proximity-type method). Vidondo & Voelkl [39] simulated epidemic outbreaks in cattle farms in Switzerland and determined that dynamic network measures are better at detecting simulated outbreaks than static network measures, similar to our finding that the Network Connectivity method informed by the two-month networks detected more outbreaks than the All-Month Network Connectivity method. Network Connectivity-type methods have also been shown to be more effective than random methods for finding outbreaks in endemic settings for other diseases (e.g., Ribeiro-Lima et al. [47] for bovine tuberculosis).

The data that we have from the Republic of Türkiye contain very detailed information on the reported outbreaks, and on shipments of cattle between January 2007 and July 2012. However, unreported outbreaks or shipments could still limit the potential for outbreak detection in our analysis. We assume that we are missing either outbreaks or cattle movements

or both because there were some outbreaks that were not detected by all three surveillance methods (Fig H in S1 Text), implying the existence of either un-reported outbreaks that could bridge the gaps between reported outbreaks or missing connections between epiunits. We also only had the exact locations of 40,746 out of 54,096 epiunits, which likely impeded the effectiveness of the Spatial Proximity method. We also did not have information about the location or shipments to or from slaughterhouses/abattoirs, markets or other non-farm hubs; such data would have been worth including in the analysis as potential surveillance targets (see Gunasekara et al. [48] and Siengsanan-Lamont et al. [49] that did explore surveillance methods based on slaughterhouses/abbatoirs, markets, etc.). However, as one is unlikely to encounter a more complete dataset from a country with endemic FMD it is important to work around these data limitations to derive findings to inform surveillance allocations in other, more resource-limited, countries with endemic FMD.

Further, we know that the Republic of Türkiye recommends certain control measures (mass vaccinations, reactive ring vaccinations, reactive movement controls) [19,33,50] but we do not know to what extent those control measures were enacted around the country from January 2007 to July 2012, so we did not factor this into our sentinel surveillance method designs. It is possible that control measures in the Republic of Türkiye managed to stop the spread of FMD outbreaks to certain epiunits between January 2007 and July 2012 that would otherwise have experienced outbreaks, which would make all of our sentinel surveillance methods appear less effective. This uncertainty may also restrict the applicability of our results to other countries with endemic FMD that have fewer control measures in place, as FMD may spread differently in them. However, it is likely that our data-informed sentinel surveillance methods would perform better in countries with no FMD control measures, because the methods we propose here do not account for the presence of control measures. Control measures known to have occurred during this time in the Republic of Türkiye, include reduced cattle shipments (reactive movement controls) and reduced transmission to nearby epiunits through vaccination after outbreak detection (reactive ring vaccination), thereby potentially directly reducing the efficacy of the Surveillance Methods we explore in this study [19,33,50]. Nonetheless, even in the presence of such control activities it remains important to identify subsequent outbreaks and we show that methods based on spatial or network properties strongly outperform random search. It is possible that, if such controls were recorded, they could be incorporated into a sentinel strategy to gain further efficiency. It would be worthwhile for future studies to use a dynamic model of both surveillance and control activities (similar to that used in [50]) to assess how the performance of surveillance strategies might interact.

In this study, we focus on using each sentinel surveillance method independently so that we can easily control the surveillance effort level and assess how well each method performs alone. However, it would be possible to develop a composite measure of outbreak risk using multiple methods at the same time (e.g., Dawson et al. [28] and Kendall et al. [51] suggest a hybrid of the Network Connectivity and Network Proximity methods) or varying the method we use over time (via an adaptive management framework, [52]). It is possible that combining surveillance methods (either across time or at one time) might also help reduce the number of outbreaks detected by no method (Fig H in S1 Text) and also allow us to survey more than 35% of epiunits consistently. In this study, to ensure direct comparison of each surveillance method, we could only explore surveillance efforts up to 35% as the Network Proximity method was unable to survey 35% of the network in certain months (Figs F and G in S1 Text). Note that when we did assess beyond 35%, the relative performance of the surveillance methods able to survey past 35% was the same until 60% surveillance effort, at 65% surveillance effort the methods that relied on network information (see Fig 1) became limited by the number of epiunits present in the two-month networks (Fig A in S1 Text). This study focused on assessing active sentinel surveillance methods for detecting outbreaks. Had we focused on other goals such as 'early detection' the methods we assessed would have been different and thus the relevance of collecting different types of data and the effectiveness of different surveillance methods would have been different.

Since all of the data-informed sentinel surveillance methods performed comparably and the most data-informed method was only marginally better and was limited in the number of candidate epiunits it detected, the results of this study

suggest that sentinel surveillance sites in countries with endemic FMD could usefully be allocated through Spatial Proximity or Network Connectivity surveillance methods (whichever would require the least additional effort to collect the requisite information). However, based on these results, if a country is able to allocate sufficient resources to ensure a high surveillance effort, or is planning to increase the effort it allocates to surveillance in the future, it would make sense to prioritize building a network-data-motivated surveillance system. In this study we present general guidelines for designing active sentinel surveillance methods that might be useful for detecting FMD outbreaks in an endemic FMD context. However we note that developing an actual FMD surveillance system for a country would require explicit, time-specific information on farming demography and animal movement networks, FMD status, existing FMD surveillance infrastructure and available resources of the country or region of interest. Significant effort is expended on surveillance efforts, so finding the most efficient locations to survey is crucial to ensure that countries do not waste effort surveying farms that are relatively unlikely to experience outbreaks, especially if such effort could be better spent on other forms of management to help reduce future FMD outbreaks.

## Supporting information

**S1 Text. Table A.** *Comparing the All-Month Network and the Average of the Two-Month Networks* – The values in the "All-Month Network" row describe the properties of the All-Month Network while the values in the second row describe the average properties of the 66 Two-Month Networks (i.e., the value in the Median Node Degree column represents the average Median Node Degree across the 66 Two-Month Networks). We computed the median value of the Node Degree and the median value of the Betweenness Centrality because the distributions of both for the All-Month network and the Two-Month Networks were fairly skewed. The IQR (Interquartile Range) columns show the 25% quantile – 75% quantile, showing the average value of the 25% quantile and of the 75% quantile in the "Average of the Two-Month Networks" row. **Fig A**. *Performance of All Surveillance Methods Across Surveillance Effort Levels* - The points represent the average percent of outbreaks each surveillance method detected across all $t+1$ months at each surveillance effort level. The ribbons represent the variability across months (the interquartile range). The Network Connectivity method is unable to survey over 60% of the epiunits for all of the two-month networks and so it is only plotted until 60% Surveillance Effort Level. The Network Proximity method is unable to survey over 30% of the epiunits for all of the two-month networks and so it is only plotted until 30% Surveillance Effort Level. **Fig B**. *Variation in Percentage of Network Searched Across Surveillance Methods and Surveillance Effort Levels* - The scatter plots show the percentage of total epiunits (54,096) searched by each of the Data-Informed Surveillance method in each $t+1$ month. **Fig C**. *Two-Month Network Connectivity Metric Performance* - (a) The percentage of outbreaks detected (on average across months) when the Network Connectivity method ('Network Connectivity (Degree)', see Fig 1 main text) is used in comparison to the percentage of outbreaks detected (on average across months) by other node-level network connectivity metrics (Table B in S1 Text) on average, across surveillance effort levels. (b) The percentage of outbreaks detected by the Network Connectivity method (Fig 1 main text) in each $t+1$ month at the 10% Surveillance Effort Level. (c) The percentage of outbreaks detected by the other node-level network connectivity metrics (Table B in S1 Text) in each $t+1$ month at the 10% Surveillance Effort Level. Note the similarity in shape between (b) and (c) even though some of the metrics used in (c) utilize the directed, weighted versions of the two-month networks and the degree metric used in (b) used only the information found in the undirected, unweighted versions of the two-month networks. (d) Matrices showing the Spearman correlation coefficients between each of the network connectivity metrics across months, at each surveillance effort level (one matrix per level). The correlation coefficients across all 7 of these matrices, across all 9 metrics, are always positive and the median correlation is 0.81. **Table B**. *Other Node-Level Network Connectivity Metrics* - The names in the first column correspond with the names used in Fig C in S1 Text to indicate the metric in question. 'Epiunit $i$' denotes the focal epiunit for whom all of these metrics are calculated. Note that self-loops were removed from the networks and there can only ever be one edge from any epiunit $i$ to any epiunit $j$ (as duplicate edges were collated in the weight of said edge). The weight of any edge corresponds to the frequency

of shipments along that path during the time period in question for that network. All of these metrics were calculated using functions (betweenness, eigen_centrality, strength, coreness, degree respectively) from the igraph package in R (Csárdi et al., 2023, v. 1.5.0).[42]. **Fig D**. *Performance of All Surveillance Methods Across Surveillance Effort Levels* - The points represent the average percent of outbreaks each surveillance method detected across all $t+1$ months at each surveillance effort level. The ribbons represent the variability across months (the interquartile range). The All-Month Network Connectivity method does not have any variability across months because it is calculated on the all-time network, which combines all of the cattle shipments across all of the months in the dataset. The Network Proximity method is unable to survey over 30% of the epiunits for all of the two-month networks and so it is only plotted until 30% Surveillance Effort Level. **Fig E**. *Survey Radius Used for the Spatial Proximity Method Across Surveillance Effort Levels and Months* – (a) Every point represents the survey radius used to survey x% of the total epiunits, for that particular $t+1$ month at that particular Surveillance Effort Level. For every epiunit we only had a latitude and longitude, so the distances were initially calculated in degree units and then converted to kilometres by approximating 1 degree = 111km. (b) Every point represents the number of outbreaks that occurred in each $t+1$ month. This is included to demonstrate the inverse relationship between the survey radius required and the number of outbreaks in each month. **Fig F.** *Month-by-Month Surveillance Method Performance* - The bars in each panel show the number of outbreaks detected by each surveillance method at each surveillance effort level (10%, 20%, 25%, 35% respectively) at each $t+1$ month. The black line shows the number of outbreaks reported in each $t+1$ month. The $t+1$ months are ordered by decreasing number of outbreaks, with the left-most $t+1$ month having the highest prevalence of outbreaks. Versions of these graphs that correspond to 5%, 15% and 30% surveillance effort levels are in Fig 3 main text. **Fig G**. *Best Surveillance Methods Each Month* - Each panel shows the percent of outbreaks detected by the data-informed surveillance method that detected the most outbreaks in each $t+1$ month at each surveillance effort level. The bars indicate the percent of outbreaks detected by the best surveillance method and the horizontal line indicates the percent of outbreaks detected by the Random surveillance method at that surveillance effort level. The coloured bars indicate the data-informed surveillance method that detected the most outbreaks at that surveillance effort level for that $t+1$ month. The $t+1$ months are ordered by declining number of outbreaks, with the left-most $t+1$ month having the highest incidence of outbreaks. Versions of these graphs that correspond to 5%, 15% and 30% surveillance effort levels are in Fig 4 main text. **Table C**. *Average Performance of Surveillance Methods Over Time Across Surveillance Effort Levels for Serotype A* - The body of the table describes the average percent of outbreaks each surveillance method detected across $t+1$ months at each surveillance effort level. Note that a few months had 0 outbreaks of Serotype A FMD. **Table D**. *Average Performance of Surveillance Methods Over Time Across Surveillance Effort Levels for Serotype O* - The body of the table describes the average percent of outbreaks each surveillance method detected across $t+1$ months at each surveillance effort level. Note that a few months had 0 outbreaks of Serotype O FMD. **Fig H**. *Outbreaks Detected by No Data-Informed Surveillance Method* - The plots show the percentage of outbreaks detected by none of the Data-Informed surveillance method in each $t+1$ month at each surveillance effort level.
(DOCX)

## Acknowledgments

We wish to thank Glen David Guyver-Fletcher, Deepit Bhatia, Kaiyue Zou and Callum Arnold for their helpful comments on this project.

## Author contributions

**Conceptualization:** Ariel Greiner, José L. Herrera-Diestra, Michael Tildesley, Katriona Shea, Matthew Ferrari.

**Data curation:** Michael Tildesley, Katriona Shea, Matthew Ferrari.

**Formal analysis:** Ariel Greiner, José L. Herrera-Diestra.

**Funding acquisition:** Michael Tildesley, Katriona Shea, Matthew Ferrari.

**Investigation:** Ariel Greiner, Matthew Ferrari.

**Methodology:** Ariel Greiner, José L. Herrera-Diestra, Michael Tildesley, Katriona Shea, Matthew Ferrari.

**Project administration:** Ariel Greiner, Michael Tildesley, Katriona Shea, Matthew Ferrari.

**Resources:** Michael Tildesley, Katriona Shea, Matthew Ferrari.

**Software:** Ariel Greiner, José L. Herrera-Diestra, Matthew Ferrari.

**Supervision:** José L. Herrera-Diestra, Michael Tildesley, Katriona Shea, Matthew Ferrari.

**Visualization:** Ariel Greiner, Katriona Shea, Matthew Ferrari.

**Writing – original draft:** Ariel Greiner.

**Writing – review & editing:** Ariel Greiner, José L. Herrera-Diestra, Michael Tildesley, Katriona Shea, Matthew Ferrari.

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
