## [Decision Letter · Decision Letter 0]

PCOMPBIOL-D-24-01341Allocating Limited Surveillance Effort for Outbreak Detection of Endemic Foot and Mouth DiseasePLOS Computational Biology Dear Dr. Greiner, Thank you for submitting your manuscript to PLOS Computational Biology. After careful consideration, we feel that it has merit but does not fully meet PLOS Computational Biology's publication criteria as it currently stands. Therefore, we invite you to submit a revised version of the manuscript that addresses the points raised during the review process. Please submit your revised manuscript within 60 days Jan 15 2025 11:59PM. If you will need more time than this to complete your revisions, please reply to this message or contact the journal office at ploscompbiol@plos.org. Please include the following items when submitting your revised manuscript: * A rebuttal letter that responds to each point raised by the editor and reviewer(s). You should upload this letter as a separate file labeled 'Response to Reviewers'. This file does not need to include responses to formatting updates and technical items listed in the 'Journal Requirements' section below.* A marked-up copy of your manuscript that highlights changes made to the original version. You should upload this as a separate file labeled 'Revised Manuscript with Track Changes'.* An unmarked version of your revised paper without tracked changes. You should upload this as a separate file labeled 'Manuscript'. If you would like to make changes to your financial disclosure, competing interests statement, or data availability statement, please make these updates within the submission form at the time of resubmission. Guidelines for resubmitting your figure files are available below the reviewer comments at the end of this letter. We look forward to receiving your revised manuscript. Kind regards, Eric HY Lau, Ph.D.Academic EditorPLOS Computational Biology Thomas LeitnerSection EditorPLOS Computational Biology Feilim Mac GabhannEditor-in-ChiefPLOS Computational Biology Jason PapinEditor-in-ChiefPLOS Computational Biology  **Journal Requirements:** **Additional Editor Comments (if provided):** The Authors are expected to address all the criticisms by all Reviewers. In particular, please assess if there were any overfitting issues when evaluating the three surveillance methods, clarify if the analysis was retrospective or prospective in nature, so the assessment is likely applicable to future outbreaks (Reviewer #1), describe the existing surveillance system in the study period, and how it informed outbreak response and control measures, reconsider how the existing surveillance would affect the comparison between surveillance methods (Reviewer #2). In additional to the above comments, please address,

1. Please provide a definition of FMD outbreak in the Republic of Türkiye.

2. Network construction, “FMD has an incubation time of 2-14 days (33) and a serial interval of 8-9 days (34), thus it could take up to twenty days for one cow to become infected with FMD and transmit it to another cow”. This seems to suggest that the surveillance system considered is symptomatic based. However it may help to describe clearly the target of the surveillance system in the Methods.

3. If the surveillance system is symptom based, were the authors considering a more intensive active surveillance on top of the existing FMD reporting? Some more background information on the existing FMD surveillance would be helpful.

4. 7 levels of surveillance effort (5%, 10%, 15%, 20%, 25%, 30%, 35%) were considered. Were these dynamic sentinel surveillance sites based on the FMD outbreak situation in the past one or two months, or is a static sentinel surveillance structure?

5. I would like to know the approximate sentinel surveillance effort in practice, and would the 7 levels above cover the practical range of available resources? For example, would surveillance effort of <5% be most practical?**Reviewers' comments:** Reviewer's Responses to Questions

**Comments to the Authors:**

Reviewer #1: The manuscript entitled “Allocating Limited Surveillance Effort for Outbreak Detection of Endemic Foot and Mouth Disease” developed three approaches to detect FMD outbreaks in Turkey. The results indicated that both “Spatial Proximity” and “Network Connectivity” can detect outbreaks with less data-intensive. This is a well-written paper which provided useful information to enhance the efficiency of FMD surveillance system. Below are my comments.

1. The study period is from 2007-2012. I was wondering why no more recent data is available for analysis. Whether the data 12 years ago is still able to represent the current situation?

2. In figure 1. The “Network Connectivity” method does not include outbreak information. How do define the outbreak has been detected by this method? Following the similar question, what’s the proportion of the outbreaks NOT detected by the three methods?

3. The analysis result indicated that data-informed surveillance methods perform better to detect FMD outbreaks with limited effort. Do the outbreaks detect by these methods are the number of the outbreaks within the two-month period or the exact locations of those outbreaks? In Lines 504-506 : “ Since the surveillance methods detected more outbreaks when outbreaks were rare, it implies that outbreaks may be easier to predict and track using data-informed methods in countries with endemic FMD when they are rare,” Another possibility is overfitting issue,

4. In Figure 2. Though the three methods detected more % of outbreaks than random approach. The slope of the curves gradually reached the plateau that indicate more efforts might not enhance the capacity well. In the real world, whether 35% of effort is enough to control FMD outbreaks?

5. Following the previous question, whether these methods can be used to predict future outbreaks? The analysis is designed to use retrospective data; however, the intervention should be adopted as early as possible to stop the transmission. If the approach is not able to predict outbreaks in advance, disease prevention may not easy to be achieved.

Reviewer #2: This paper presents a study where retrospective analysis of FMD outbreaks and cattle movements is used to estimate and compare the performance of different surveillance strategies. It is based on data on FMD outbreaks and cattle shipments between epiunits in a country with endemic FMD (Turkey). The authors conclude that all data-informed approaches are better than random sampling.

General comments

This is nice initiative to retrospectively analyse outbreak data from the field. However, the study does not take into account the preventive and disease control measures applied after the detection of outbreaks. The discussion and conclusions completely disregard the fact that the number of outbreaks have been influenced by disease control measures based on the same principles as the simulated surveillance strategies (spatial proximity and network proximity).

Language and structure

The language is well-written and easy to read. However, there is some repetition that could be removed. For example, the surveillance methods are very clearly described in the Material and methods section. Please consider removing further explanations in other sections of the manuscript.

Also, the structure of the manuscript is not as stringent as you would expect from a scientific paper. E.g. main conclusions are included in the introduction.

Main objections

When an outbreak is detected, strong control measures are usually implemented in a zone around the infected unit(s). If applied correctly, such immediate measures would decrease disease spread in the geographical proximity of farms. As a consequence, simulated spatial proximity surveillance on data from such a reality would seem less efficient. Imagine not taking any spatial proximity-based actions after an outbreak is detected. I think retrospective analysis would then show many new outbreaks based on spatial proximity surveillance. In other words, the performance of the strategies and which one performs best really depends on the measures taken. Many countries would apply a zone and then do some kind of contact tracing. To do risk-based surveillance for FMD and take measures based on network connectivity only is probably the least common strategy. And if that’s not applied, then of course, in a study like this, some additional cases will be found through that strategy. And this strategy will seem slightly better. But again, it all depends on what control measures and additional surveillance were applied in the region. It is important that these aspects are considered. Please rethink and expand on this and adjust parts of your discussion and conclusions accordingly.

It would be helpful to include a bit more about the Turkish strategy/action plan to better understand what measures they apply once an outbreak is detected.

FMD is a disease with strong clinical symptoms and therefore, passive clinical surveillance can be expected to be relatively sensitive. Surveillance components are usually part of a surveillance system and when we assess them it is important to also consider other components of the system. Please add a discussion about how the performance of these active surveillance strategies compare to passive clinical surveillance.

Some detailed comments

Rows 137-143. A summary of the results and conclusions are given already at the end of the Introduction. Usually, we expect this in an abstract, not the introduction. Please remove.

Row 209: It is common practice to add active surveillance to existing passive clinical surveillance, and it is also common to combine active surveillance efforts. For example, establishing a zone for surveillance and disease control is usually combine with contact tracing (as you also describe in the paper). What was the reason you did not consider combinations of these different approaches? Based on my experience, in reality, the most optimal allocation would be a combination more than one method. Please consider adding such calculations to your study or clarify why this is not included.

Rows 263-266. I suggest removing this unrealistic scenario from the study. It is indeed unrealistic and does not add anything significant to the conclusions.

Row 280. What radiuses did this strategy result in? This is of interest to everyone who are used to zone sizes that are commonly practiced in disease control and surveillance.

Rows 354-366. Did you test statistical differences between the different strategies? If so, how? When you say that they were similar or one strategy detected fewer outbreaks than another, how do we know this is a true difference or not?

Fig. 2. Colours cannot be seen when the paper is printed in black and white. Thereby it is impossible to distinguish the line/methods in the graph. I suggest adding different shapes to the points of the four lines. It would also help to see the interquartile ranges more clearly.

Rows 368-372: How is this result of relevance? Please clarify or remove.

**Have the authors made all data and (if applicable) computational code underlying the findings in their manuscript fully available?**

Reviewer #1: Yes

Reviewer #2: **No: ** They have made available what they can, but the original data has to be requested from Turkish authorities.

PLOS authors have the option to publish the peer review history of their article (what does this mean? ). If published, this will include your full peer review and any attached files.

**Do you want your identity to be public for this peer review?** For information about this choice, including consent withdrawal, please see our Privacy Policy .

Reviewer #1: No

Reviewer #2: No

 **Figure resubmission:**While revising your submission, please upload your figure files to the Preflight Analysis and Conversion Engine (PACE) digital diagnostic tool, https://pacev2.apexcovantage.com/. PACE helps ensure that figures meet PLOS requirements. To use PACE, you must first register as a user. Registration is free. Then, login and navigate to the UPLOAD tab, where you will find detailed instructions on how to use the tool. If you encounter any issues or have any questions when using PACE, please email PLOS at figures@plos.org. Please note that Supporting Information files do not need this step. If there are other versions of figure files still present in your submission file inventory at resubmission, please replace them with the PACE-processed versions. 
---

## [Decision Letter · Decision Letter 1]

PCOMPBIOL-D-24-01341R1

Allocating Limited Surveillance Effort for Outbreak Detection of Endemic Foot and Mouth Disease

PLOS Computational Biology

Dear Dr. Greiner,

Thank you for submitting your manuscript to PLOS Computational Biology. After careful consideration, we feel that it has merit but does not fully meet PLOS Computational Biology's publication criteria as it currently stands. Therefore, we invite you to submit a revised version of the manuscript that addresses the points raised during the review process.

Please submit your revised manuscript within 30 days Jun 22 2025 11:59PM. If you will need more time than this to complete your revisions, please reply to this message or contact the journal office at ploscompbiol@plos.org. Please include the following items when submitting your revised manuscript:

We look forward to receiving your revised manuscript.

Kind regards,

Eric HY Lau, Ph.D.

Academic Editor

PLOS Computational Biology

Thomas Leitner

Section Editor

PLOS Computational Biology

**Additional Editor Comments :**

The Authors have addressed most of the comments raised by the editor and reviewers. Please address further comments by Reviewer #3, in particular provide more discussion on the practical considerations around the main findings.

**Journal Requirements:**

1) Please provide an Author Summary. This should appear in your manuscript between the Abstract (if applicable) and the Introduction, and should be 150-200 words long. The aim should be to make your findings accessible to a wide audience that includes both scientists and non-scientists. Sample summaries can be found on our website under Submission Guidelines:

3) Please update your Data Availability Statement in the online submission form.

**Reviewers' comments:**

Reviewer's Responses to Questions

**Comments to the Authors:**

**Please note that one of the reviews is uploaded as an attachment.**

Reviewer #1: The authors have respond all the comments appropriately.

Reviewer #3: Review comments are in the uploaded attachment

**Have the authors made all data and (if applicable) computational code underlying the findings in their manuscript fully available?**

Reviewer #1: None

Reviewer #3: Yes

PLOS authors have the option to publish the peer review history of their article (what does this mean? ). If published, this will include your full peer review and any attached files.

**Do you want your identity to be public for this peer review?** For information about this choice, including consent withdrawal, please see our Privacy Policy .

Reviewer #1: No

Reviewer #3: **Yes: ** Susanna Sternberg Lewerin

**Figure resubmission:**
---

## [Editor Report · Decision Letter 2]

Dear Dr. Greiner,

We are pleased to inform you that your manuscript 'Allocating Limited Surveillance Effort for Outbreak Detection of Endemic Foot and Mouth Disease' has been provisionally accepted for publication in PLOS Computational Biology.

Best regards,

Eric HY Lau, Ph.D.

Academic Editor

PLOS Computational Biology

Thomas Leitner

Section Editor

PLOS Computational Biology

Thanks for addressing all the editor’s and reviewers' comments. Congratulations on the excellent work!

---

## [Editor Report · Acceptance letter]

PCOMPBIOL-D-24-01341R2

Allocating Limited Surveillance Effort for Outbreak Detection of Endemic Foot and Mouth Disease

Dear Dr Greiner,

I am pleased to inform you that your manuscript has been formally accepted for publication in PLOS Computational Biology. Your manuscript is now with our production department and you will be notified of the publication date in due course.

With kind regards,

Judit Kozma
